# The Association between 4-Tertiary-Octylphenol, Apoptotic Microparticles, and Carotid Intima-Media Thickness in a Young Taiwanese Population

**DOI:** 10.3390/toxics11090757

**Published:** 2023-09-06

**Authors:** Chien-Yu Lin, Ching-Way Chen, Chikang Wang, Fung-Chang Sung, Ta-Chen Su

**Affiliations:** 1Department of Internal Medicine, En Chu Kong Hospital, New Taipei City 237, Taiwan; 00724@km.eck.org.tw; 2School of Medicine, Fu Jen Catholic University, New Taipei City 242, Taiwan; 3Department of Environmental Engineering and Health, Yuanpei University of Medical Technology, Hsinchu 300, Taiwan; ckwang@mail.ypu.edu.tw; 4Department of Cardiology, National Taiwan University Hospital Yunlin Branch, Yunlin 640, Taiwan; y05213@ms1.ylh.gov.tw; 5Department of Health Services Administration, China Medical University College of Public Health, Taichung 404, Taiwan; fcsung@mail.cmu.edu.tw; 6Department of Food Nutrition and Health Biotechnology, Asia University, Taichung 413, Taiwan; 7Department of Environmental and Occupational Medicine, National Taiwan University Hospital, Taipei 100, Taiwan; 8Division of Cardiology, Department of Internal Medicine, National Taiwan University Hospital, Taipei 100, Taiwan; 9Institute of Environmental and Occupational Health Sciences, College of Public Health, National Taiwan University, Taipei 100, Taiwan

**Keywords:** 4-tertiary-octylphenol (4-tOP), apoptotic microparticles, CD31+/CD42a−, CD31+/CD42a+, common carotid artery intima-media thickness (CIMT)

## Abstract

As one of the most common alkylphenols, 4-tertiary-octylphenol (4-tOP) is commonly used in many consumer products. Our previous epidemiological study revealed a negative correlation between serum 4-tOP levels and carotid intima-media thickness (CIMT), which serves as a biomarker of arteriosclerosis. We aimed to explore the role of apoptotic microparticles, markers of vascular endothelial cell function, in the 4-tOP and CIMT connection. To investigate this, we enrolled 886 Taiwanese adolescents and young adults (aged 12–30 years) and examined the relationships among serum 4-tOP levels, apoptotic microparticles (CD31+/CD42a−, CD31+/CD42a+), and CIMT. Our results showed negative associations among serum 4-tOP levels, both apoptotic microparticles, and CIMT in multiple linear regression analysis. The odds ratios for CIMT (≥75th percentile) and the natural logarithm of 4-tOP were highest when both CD31+/CD42a− and CD31+/CD42a+ were greater than the 50th percentile. Conversely, the odds ratios were lowest when both CD31+/CD42a− and CD31+/CD42a+ were less than the 50th percentile. In the structural equation model, we demonstrated that serum 4-tOP levels were negatively correlated with CIMT and indirectly and negatively correlated with CIMT through both apoptotic microparticles. In conclusion, our study reported the inverse association between 4-tOP apoptotic microparticles and CIMT in a young Taiwanese population. Further experimental studies are needed to clarify these associations.

## 1. Introduction

Endocrine-disrupting substances are environmental pollutants that can alter reproductive and developmental processes in living organisms [1]. Alkylphenols are a group of man-made chemicals that are obtained via the alkylation of phenols. They are classified as endocrine-disrupting substances due to their estrogenic effects [2]. 4-tertiary-octylphenol (4-tOP) is an alkylphenol that is frequently used in the manufacture of adhesives, cleaners, and polymers, such as epoxy resins and polycarbonates [3]. Due to its hazardous nature and wide application, the European Union regulates 4-tOP [4]. However, 4-tOP is still released into the environment by other countries and is not routinely monitored [5]. Recent reports have found a higher level of alkylphenols in the rivers of Taiwan [6]; the detection rate of 4-tOP was 100% in a Taiwanese cohort [7]. Therefore, it is crucial to investigate the potential health consequences of 4-tOP for the Taiwanese population.

As an estrogen receptor agonist, 4-tOP exposure was found to be associated with toxic effects on developmental and reproductive systems in animal studies [8,9], as well as in observational studies [10,11]. Additionally, exposure to other alkylphenols, such as bisphenol A, has been correlated with cardiovascular disease (CVD) [12]. Reports concerning 4-tOP and its effect on the cardiovascular system have varied in consistency, particularly in in vitro and animal studies. In isolated rat islet cells, 4-tOP treatment has been found to change the morphology and function of *β*-cells [13]. Moreover, increased oxidative stress and apoptosis of kidney tissue were observed in 4-tOP-exposed male rats [14]. However, 4-tOP could protect pancreatic islets from drug-induced apoptosis in diabetic mice [15]. Additionally, acute exposure to 4-tOP evoked a rapid relaxation of vascular smooth muscle cells [16]. Decreased body weight and reduced fat in adipocytes were also observed in pregnant rats treated with 4-tOP [17]. In epidemiology, few reports have investigated the relationships between 4-tOP and CVD and its risk factors. One study explored the impact of environmental pollutants on maternal metabolism in pregnant Chinese women and found that 4-tOP induced significant metabolomic changes involved in lipid and carbohydrate metabolism [18]. In our previous study, we showed that serum 4-tOP concentrations were negatively correlated with CVD risk factors and carotid intima-media thickness (CIMT), a marker of subclinical arteriosclerosis [7]. Nevertheless, no epidemiological investigations to date have explored the potential protective mechanism of 4-tOP in CVD.

Endothelial cell apoptosis, triggered by both external and internal events, plays an important role in arteriosclerosis [19]. Microparticles are fragments of the cell membrane (100–1000 nm) shed from cells [20]. Apoptotic microparticles are released during the apoptotic process and are not only biomarkers of endothelial cell dysfunction, but also contribute to the development of pathological conditions [21,22]. Apoptotic vascular endothelial cells release CD31+/CD42a−, whereas CD31+/CD42a+ is excreted from apoptotic platelets [23,24]. One recent epidemiological study revealed a negative correlation between estrogen levels and procoagulant microparticles in newly menopausal women [25]. Moreover, endocrine-disrupting substance exposure has been linked to apoptotic microparticles in previous reports [24,26]. However, existing reports have not examined the role of 4-tOP in vascular endothelial cell function, nor the connections between 4-tOP exposure, apoptotic microparticles, and subclinical arteriosclerosis. The goal of this research was to explore the relationships among serum 4-tOP concentrations, apoptotic microparticles (CD31+/CD42a− and CD31+/CD42a+), and CIMT in a cohort composed of a young Taiwanese population.

## 2. Materials and Methods

### 2.1. Study Population and Data Collection

We established a cohort study (YOTA, Young Taiwanese Cohort Study) of 886 young Taiwanese people (12–30 years old) selected from a Taiwanese urine screening project from 2006 to 2008 [27]. This study was approved by the National Taiwan University Hospital Research Ethics Committee. The study participants were enrolled in this study after informed consent was obtained. We confirmed that all methods were performed in accordance with the relevant guidelines and regulations. All 886 subjects participated in this study. Details of the study population are provided in the Appendix A Study population and data collection.

### 2.2. Measurement of Serum 4-tOP Levels

4-tOP standard was acquired from Sigma/Aldrich (St. Louis, MO, USA), while 4-n-Octyl-d17-phenol was provided by C/D/N Isotopes (Pointe-Claire, Montreal, QC, Canada), boasting a purity exceeding 98%. Preparations of stock solutions for each compound were made at a concentration of 500 µg/mL in methanol and securely stored at −20 °C. Milli-Q water, acquired via a Millipore water purification system (Milford, MA, USA), was employed. N-methylmorpholine (with a purity surpassing 99.5%) was supplied by J.T. Baker (Phillipsburg, NJ, USA). Bovine plasma was procured from Sigma-Al-drich (St. Louis, MO, USA) and kept at −80 °C.

To analyze 4-tOP concentrations, serum samples were employed using the method previously described [28]. The experiment was conducted using glassware cleaned with methanol. Serum samples underwent preparation via protein precipitation. The quantification of 4-tOP was achieved using an ultra-performance liquid chromatography tandem mass spectrometer. Before the instrumental analysis was conducted, both samples and matrix-matched standard calibration solutions were meticulously prepared. These calibration solutions were created by combining bovine serum with varying concentrations of spiked 4-tOP (10, 25, and 100 ng/mL, respectively), acting as internal standards for analysis. The calibration curves exhibited linear ranges spanning 2.5–500 ng/mL for 4-tOP, encompassing 8 data points. The analysis consistently demonstrated a coefficient of determination equal to or exceeding 0.997. The limit of quantitation for 4-tOP was determined to be 0.9 ng/mL. In terms of precision, the intra-assay and inter-assay coefficients of variation were established at 5% and 10%, respectively.

### 2.3. Measurement of Apoptotic Microparticles

Apoptotic microparticles in the serum were quantified using a flow cytometer. The analysis involved assessing microparticles in citrated serum using the following fluorescent monoclonal antibodies: phycoerythrin-labeled anti-CD31 (from BD Bioscience), fluorescein isothiocyanate-labeled anti-CD42a (from BD Bioscience), and fluorescein isothiocyanate-labeled anti-CD14 (from BD Bioscience). The resulting microparticle values are presented as counts per microliter (µL).

### 2.4. Measurement of CIMT

The assessment of CIMT involved measuring the distance between two echogenic lines on the far wall of the vessel. The first echogenic line marked the boundary between the vessel’s lumen and its inner lining, while the second echogenic line signified the separation between the middle layer of the vessel and its outer layer. A skilled technician employed a high-resolution *B*-mode ultrasonography system (GE Vivid 7 ultrasound system, Horten, Norway) with a real-time 3.5–10 MHz *B*-mode scanner to examine the CIMT of extracranial carotid arteries. To determine CIMT, a specialized software package for vascular ultrasound was utilized for automatic calculations after the examination. The comprehensive details can be found in the Appendix A Measurement of CIMT.

### 2.5. Covariates

Covariates for this study included: age (continuous), gender (categorial), household income (categorial), hypertension (categorial), diabetes mellitus (categorial), smoking status (categorial), alcohol consumption (categorial), z score of body mass index (BMI) (continuous), systolic blood pressure (SBP) (continuous), and low-density lipoprotein cholesterol (LDL-C) (continuous), high-density lipoprotein cholesterol (HDL-C) (continuous), triglyceride (continuous), uric acid (continuous), homeostasis model assessment of insulin resistance index (HOMA-IR) (continuous). The details are available in the Appendix A Covariates.

### 2.6. Statistical Analysis

The mean concentrations of 4-tOP, apoptotic microparticles, and CIMT were expressed in subpopulations and analyzed using a two-tailed Student’s *t*-test. The relationships of 4-tOP and microparticles with CVD risk factors and CIMT were analyzed via multiple linear regression models. The model was adjusted for age, sex, and smoking and drinking status (Model 1). To further study the relationships between the 4-tOP level, microparticles, and CIMT, we adjusted for Model 2, which consisted of Model 1 plus the BMI z score, SBP, LDL-C, HDL-C, triglyceride, and HOMA-IR as confounders. Via logistic regression analysis, we also analyzed the odds ratios associated with thicker CIMT (≥75th percentile) with a natural log-transformed unit increase in 4-tOP concentration in different apoptotic microparticle subgroups.

Structural equation modeling (SEM) was applied to verify the role of apoptotic microparticles in the association between 4-tOP and CIMT. Covariate factors (Model 2) were also integrated into the SEM analysis. SPSS AMOS was used to estimate the parameters of the SEM analysis. Two SEM analyses were performed for two apoptotic microparticles: CD31+/CD42+ and CD31+/CD42−. Natural log transformations were performed for 4-tOP, triglyceride, and HOMA-IR due to deviations from a normal distribution. In this study, a *p* value of < 0.05 was considered significant.

## 3. Results

The range for serum 4-tOP was 102.1 ng/mL (from a minimum of 2.7 ng/mL to a maximum of 104.8 ng/mL), with a detection rate of 100% in this cohort. Table 1 shows the mean and standard deviation of serum 4-tOP, apoptotic microparticles, and CIMT in different population subgroups. The study consisted of 350 male and 536 female subjects. 4-tOP levels were higher in the older age group (20–30 years old) but were not different between other subpopulations. Serum CD31+/CD42a− counts were higher in the younger age group and in subjects with higher BMI z scores, hypertension, and diabetes mellitus, while CD31+/CD42a+ counts were higher in the younger age group. In addition, CIMT was thicker in male subjects and those with higher BMI z scores, alcohol consumption, hypertension, and diabetes mellitus.

Table 2 shows the associations between a 1-unit increase in ln 4-tOP/apoptotic microparticles and CVD risk factors/CIMT. We reported that measures of BMI z score, HOMA-IR, and CIMT decreased significantly with an elevation of 4-tOP levels, whereas HDL-C was positively associated with 4-tOP. There were positive relationships between CD31+/CD42a− counts and all CVD risk factors/CIMT except UA, while CD31+/CD42a+ counts were positively associated with BMI z score, triglyceride, HOMA-IR, and CIMT. The associations of 4-tOP with apoptotic microparticles and CIMT are shown in Table 3. Both Models 1 and 2 had similar results. Serum concentrations of 4-tOP were negatively correlated with both apoptotic microparticles and CIMT.

Table 4 demonstrates odds ratios between CIMT (≥75th percentile) and ln 4-tOP in different categories of apoptotic microparticle levels. The odds ratios were highest when both CD31+/CD42a−and CD31+/CD42a+ were greater than the 50th percentile (OR 0.553 [95% C.I. 0.340–0.901], *p* < 0.017) and lowest when both CD31+/CD42a− and CD31+/CD42a+ were less than the 50th percentile (OR 0.048 [95% C.I. 0.015–0.232], *p* < 0.001). The relationships between 4-tOP, apoptotic microparticles, and CIMT in subpopulations are demonstrated in Appendix A. Ln 4-tOP was negatively correlated with apoptotic microparticles and CIMT in all subpopulations.

The association of 4-tOP, apoptotic microparticles, and CIMT in SEMs is shown in Figure 1. For CD31+/CD42- microparticles, the SEM analysis showed that 4-tOP was negatively associated with CD31+/CD42- (B12 = −0.662, *p* < 0.001) and that CD31+/CD42- was positively associated with CIMT (B23 = 9.090, *p* < 0.001). For microparticles CD31+/CD42+, the SEM analysis showed similar results. 4-tOP was negatively associated with CD31+/CD42+ (B12 = −0.604, *p* < 0.001), and CD31+/CD42+ was positively associated with CIMT (B23 = 3.685, *p* < 0.001). The SEM analysis also showed that 4-tOP was negatively associated with CIMT (B13= −25.223, *p* < 0.001). The GFI and NFI were both one in both models. The RMS for CD31+/CD42+ was 0.001, and the RMS for CD31+/CD42- was 0.000. All of these parameters indicated perfect goodness of fit for both models. The parameters of the confounding factors for both SEM analyses are listed in Appendix A.

## 4. Discussion

In this current study, we showed an inverse correlation between serum 4-tOP levels, apoptotic microparticles (CD31+/CD42a− and CD31+/CD42a+), and CIMT. Additionally, the negative association between 4-tOP and CIMT was most evident when both apoptotic microparticles were less than the 50th percentile. The SEM analysis revealed a negative direct correlation between serum 4-tOP and CIMT, as well as a negative indirect correlation between serum 4-tOP and CIMT via apoptotic microparticle effects. This epidemiological report is the first to link serum 4-tOP, apoptotic microparticles, and subclinical atherosclerosis. Nevertheless, it is important to highlight that 4-tOP has been linked to negative outcomes on developmental and reproductive systems in both animal studies and epidemiological research [8,9,10,11]. Given its role as an estrogen receptor agonist, the effects of 4-tOP on the human body might be intricate, possibly posing risks to specific systems while impacting other aspects of health. It is crucial to underscore that our study solely involved observation and cross-sectional analysis. Consequently, it would be premature to speculate about the clinical implications, especially given the current inadequacy of robust evidence. Therefore, further research is necessary before any conclusions can be made.

Estrogen can affect cardiovascular health through direct effects on cells or indirectly through effects on systemic systems [29]. Recent research has suggested that decreased estrogen levels increased the risk of CVD in women [30]. Some endocrine-disrupting substances have been shown to associate with several CVD risk factors, including altered lipid profiles and glucose homeostasis [31]. Since 4-tOP is a xenoestrogen, several reports have explored its association with several CVD risk factors. In animal studies, high-dose 4-tOP can reduce fat deposition in fat cells, decrease body weight [17], protect pancreatic islets from drug-induced apoptosis [15], and alter cholesterol metabolism [32]. As well as the effects on CVD risk factors, one animal study also investigated the direct effects of 4-tOP on the cardiovascular system. 4-tOP exposure can block L-type calcium channels in rat cardiac and aortic smooth muscle cells and induce vessel relaxation, an effect that mimics that of estrogen [16]. In epidemiology, few studies have investigated the relationship between 4-tOP exposure and CVD risk factors. Our previous report showed that serum 4-tOP levels were inversely correlated with insulin resistance, BMI, and CIMT, although positively correlated with favorable lipids (HDL-C, apolipoprotein A1) [7]. It remains unclear if these associations were causal. It is possible that low-dose 4-tOP exposure might have beneficial effects on CVD risk factors and arteriosclerosis.

The main finding of this study is that 4-tOP levels are negatively associated with endothelial cell and/or platelet apoptotic microparticles, and possibly improve arteriosclerosis. Apoptotic microparticles, which are small membrane-bound vesicular bodies derived from apoptotic cells, can disrupt nitric oxide production and alter endothelial cell function. Moreover, apoptotic microparticles can transfer cellular content which could promote the development of arteriosclerosis [33]. Estrogen is also an important factor in maintaining and repairing the endothelium [34]. Endogenous estrogen can improve anti-inflammatory and antithrombotic properties in endothelial cells, increase nitric oxide production, reduce reactive oxygen species production, and inhibit the expression of procoagulant molecules [25]. One previous epidemiological study showed that estrogen levels were negatively correlated with procoagulant microparticles in newly menopausal women [25]. Additionally, other endocrine-disrupting substances have also been linked to apoptotic microparticles [24,35]. It is possible that 4-tOP acts as a xenoestrogen, reduces the production of apoptotic microparticles, and has a favorable effect on the endothelium.

Recent evidence suggests that alterations in gut microbial composition may be closely related to the development of CVD [36]. Therapeutic interventions that restore the intestinal barrier and improve the gut microbiota may prevent CVD [37]. Conversely, a study of obese Mexican children and adolescents showed that changes in the gut microbiota were associated with increased levels of CRP, decreased adiponectin levels, and increased endothelial dysfunction markers [38]. Dietary phenolic compounds have antioxidant and anti-inflammatory properties and some phenolic compounds may improve gut microbiota diversity and modulate the immune system [39]. Octylphenol could alter gene expression related to lipid metabolism and alter the composition of the gut microbial community in amphibians [40]. However, in a study using data from the US National Health and Nutrition Examination Survey 2005–2010, 4-tOP levels were significantly associated with ulcerative colitis [41], and alteration of the gastrointestinal microflora may play a role in the pathophysiological mechanism [42]. The results of the above study did not support our findings and we cannot speculate because this study did not show any evidence regarding intestinal microbiota. The effect of 4-tOP on the gut microbiota and whether changes in the gut microbiome play a role in the relationship between 4-tOP exposure and cardiovascular health need further investigation.

The limitations of this study are listed as follows. First, a cross-sectional study cannot conclude any causal inference. Second, our study cohort comprised a young population with abnormal blood pressure and urinalysis; we cannot apply the conclusions to the general population. Finally, we did not consider other endocrine-disrupting substances that might be co-exposed to 4-tOP.

## 5. Conclusions

We showed higher 4-tOP concentrations were inversely associated with apoptotic microparticles (CD31+/CD42a− and CD31+/CD42a+) and CIMT. We also found that the two apoptotic microparticles had a synergistic effect on the relationship between 4-tOP and CIMT. Additionally, the two apoptotic microparticles mediated the association between 4-tOP and CIMT in the SEM. However, our study relies solely on cross-sectional analysis. Consequently, speculating about the clinical implications would be premature, especially in light of the current absence of reliable evidence. Future efforts are needed to further investigate the mechanism of 4-tOP in cardiovascular health.

## Figures and Tables

**Figure 1 toxics-11-00757-f001:**
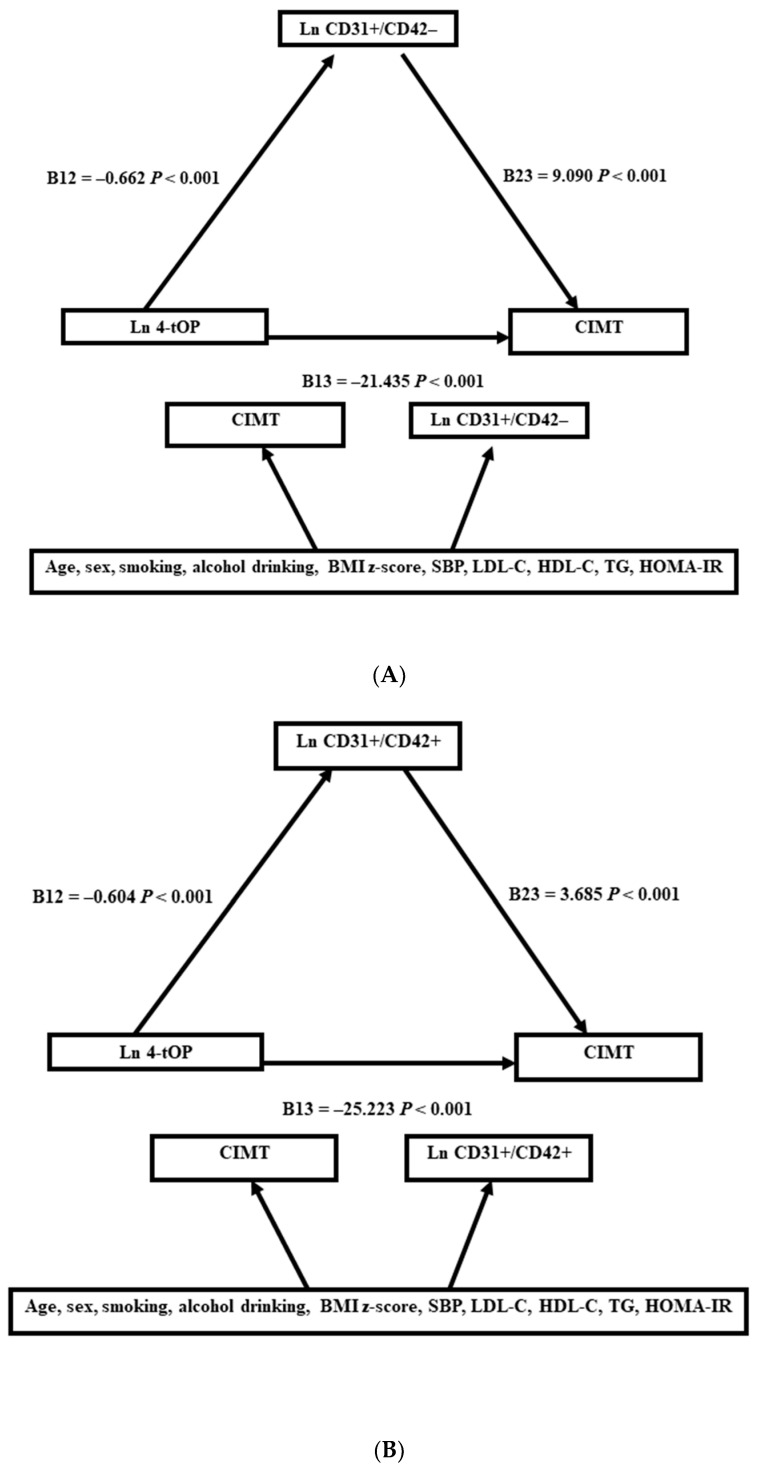
The relationship between 4-tOP, microparticles (CD32+/CD42−, CD32+/CD42+), and CIMT in the structural equation model. The covariates were adjusted as shown in Model 2. (**A**): CD32+/CD42−. (**B**): CD32+/CD42+.

**Table 1 toxics-11-00757-t001:** Basic demographics of the sample subjects, including the mean (SD) of serum 4-tOP, serum apoptotic microparticle concentrations, and CIMT.

		4-tOP (ng/mL)	CD31+/CD42a− (counts/µL)	CD31+/CD42a+ (counts/µL)		CIMT (µm)
	*n* (%)	Mean (SD)	Mean (SD)	Mean (SD)	*n* (%)	Mean (SD)
Total	886 (100)	36.9 (17.3)	369.6 (697.7)	11,261.3 (18,448.2)	883 (100)	448.2 (54.0)
Gender						
Female	536 (60.5)	36.5 (16.5)	341.7 (742.2)	11,346.4 (18,921.1)	533 (60.4)	441.5 (51.1) ^‡^
Male	350 (39.5)	37.4 (18.4)	412.3 (622.2)	11,205.7 (18,151.0)	350 (39.6)	458.3 (56.6) ^‡^
Age (years)						
12–19	275 (31.0)	33.9 (16.4) ^‡^	453.3 (674.6) *	13,545.8 (18,713.4) *	275 (31.1)	445.8 (51.9)
20–30	611 (69.0)	38.2 (17.5) ^‡^	332.0 (704.9) *	10,259.6 (18,236.6) *	608 (68.9)	449.2 (54.9)
Household income						
<50,000 TWD/per month	342 (38.7)	37.8 (16.7)	363.4 (646.2)	11,819.3 (19,177.1)	341 (38.7)	449.2 (56.3)
≥50,000 TWD/per month	542 (61.3)	36.3 (17.6)	374.1 (729.6)	10,943.4 (18,020.7)	540 (61.3)	447.3 (52.5)
BMI z score						
≤−0.19	443 (49.9)	37.0 (16.8)	293.6 (715.2) ^‡^	10,445.4 (17,197.5)	442 (50.0)	438.3 (46.5) ^‡^
>−0.19	443 (50.1)	36.7 (17.8)	445.5 (672.1) ^‡^	12,075.2 (19,603.2)	441 (50.0)	458.1 (59.0) ^‡^
Smoking status						
Active smoker	109 (12.4)	39.5 (18.5)	438.6 (783.3)	10,871.0 (16437.0)	109 (12.4)	454.6 (52.9)
Inactive smoker	776 (87.6)	36.5 (17.1)	379.2 (715.2)	11,580.0 (18,702.6)	773 (87.6)	447.2 (60.8)
Current drinking						
No	807 (91.2)	36.8 (17.1)	369.5 (711.2)	11,021.1 (18,065.9)	804 (91.2)	446.8 (53.4) *
Yes	78 (8.8)	37.3 (19.3)	373.5 (552.2)	13,749.2 (22,004.4)	78 (8.8)	462.1 (57.9) *
Hypertension						
Yes	66 (7.4)	34.1 (17.3)	681.7 (937.3) ^‡^	14,680.6 (26,740.2)	66 (7.5)	476.7 (71.8) ^‡^
No	820 (92.6)	37.1 (16.1)	343.9 (668.5) ^‡^	10,979.2 (17,585.6)	817 (92.5)	445.9 (51.6) ^‡^
Diabetes Mellitus						
Yes	17 (1.9)	36.6 (16.4)	1082.1 (1393.3) ^‡^	10,724.1 (13,747.7)	17 (1.9)	486.0 (89.9) ^‡^
No	869 (98.1)	36.9 (17.3)	356.0 (671.8) ^‡^	11,271.5 (18,532.5)	866 (98.1)	447.4 (52.8) ^‡^

*: *p* < 0.05; ^‡^: *p* < 0.005 determined via *t* test; BMI: body mass index; TWD: Taiwan dollars.

**Table 2 toxics-11-00757-t002:** Linear regression coefficients (S.E.) of cardiovascular risk factors with a unit increase in ln 4-tOP and apoptotic microparticles in multiple linear regression models (*n* = 885).

	SBP (mm Hg)	BMI z Score (kg/m^2^)	LDL-C (mg/dL)	HDL-C (mg/dL)	Ln Triglyceride (mg/dL)	Uric Acid (mg/dL)	Ln HOMA-IR	CIMT * (µm)
Ln 4-t-OP	−0.549 (0.809)	−0.234 (0.065)	2.135 (1.924)	2.453 (0.568)	0.038 (0.029)	0.030 (0.073)	−0.305 (0.059)	−30.684 (3.190)
*p* value	0.497	<0.001	0.267	<0.001	0.192	0.678	<0.001	<0.001
Ln CD31+/CD42a−	1.074 (0.336)	0.173 (0.027)	2.783 (0.799)	−1.019 (0.238)	0.069 (0.012)	0.040 (0.031)	0.203 (0.024)	14.129 (1.321)
*p* value	0.001	<0.001	0.001	<0.001	<0.001	0.193	<0.001	<0.001
Ln CD31+/CD42a+	0.194 (0.253)	0.046 (0.020)	0.668 (0.601)	0.056 (0.179)	−0.022 (0.009)	0.006 (0.023)	0.130 (0.018)	6.757 (1.024)
*p* value	0.443	0.024	0.266	0.756	0.017	0.793	<0.001	<0.001

*: *n* = 882. Adjusted for Model 1: gender, age, smoking and drinking status. Abbreviations: BMI—body mass index; HOMA-IR—homeostasis model assessment of insulin resistance; LDL-C—low-density lipoprotein cholesterol; UA—uric acid; SBP—systolic blood pressure.

**Table 3 toxics-11-00757-t003:** Linear regression coefficients (S.E.) of CIMT, apoptotic microparticles with a 1-unit increase in ln 4-tOP levels in multiple linear regression models.

	ln 4-tOP (ng/mL)				
		Model 1		Model 2	
	*n*	Adjusted *β* (S.E.)	*p* Value	Adjusted *β* (S.E.)	*p* Value
Ln CD31+/CD42a−	884	−0.736 (0.077)	<0.001	−0.672 (0.076)	<0.001
Ln CD31+/CD42a+	884	−0.730 (0.105)	<0.001	−0.609 (0.105)	<0.001
CIMT (µm)	882	−30.864 (3.190)	<0.001	−28.394 (3.142)	<0.001

Model 1 was adjusted for sex, age, smoking and drinking status. Model 2 was adjusted in the same way as Model 1, plus BMI z score, SBP, LDL-C, HDL-C, triglyceride, and HOMA-IR. Abbreviations: CIMT—carotid intima-media thickness.

**Table 4 toxics-11-00757-t004:** Odds ratios (95% confidence interval [C.I.]) of thicker CIMT (greater than the 75th percentile) with a one-unit increase in serum ln 4-tOP concentrations via different categories of apoptotic microparticle concentrations.

	No.	Odds Ratio	95% C.I.		*p* Value
			Lower	Upper	
Total	882	0.243	0.165	0.358	<0.001
CD31+/CD42a− ≤ 50%ile	441	0.095	0.037	0.241	<0.001
CD31+/CD42a− > 50%ile	441	0.420	0.270	0.654	<0.001
CD31+/CD42a+ ≤ 50%ile	441	0.057	0.023	0.139	0.009
CD31+/CD42a+ > 50%ile	441	0.425	0.272	0.663	<0.001
CD31+/CD42a− ≤ 50%ile and CD31+/CD42a+ ≤ 50%ile	319	0.048	0.015	0.232	<0.001
CD31+/CD42a− > 50%ile and CD31+/CD42a+ > 50%ile	319	0.553	0.340	0.901	0.017

Adjusted as in Model 2. Abbreviations: CIMT—carotid intima-media thickness.

## Data Availability

The datasets generated during and/or analyzed in the current study are not accessible due to privacy or ethical restrictions. Nevertheless, they can be obtained from the corresponding author with the permission of National Taiwan University Hospital upon a reasonable request.

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
