# Peer review of "The Association between 4-Tertiary-Octylphenol, Apoptotic Microparticles, and Carotid Intima-Media Thickness in a Young Taiwanese Population"

_toxics, 2023, doi:10.3390/toxics11090757_

Round 1
Reviewer 1 Report
The manuscript by Chien-Yu Lin et al, reports the role of 4-tOP in attenuating endothelial cell apoptosis which is linked to arteriosclerosis. In this epidemiological study, the authors screened the serum 4-tOP levels in the Taiwanese population. The authors have previously reported that 4-tOP levels are negatively correlated to carotid intima-media thickness (CIMT) which is a biomarker of arteriosclerosis, published in Environ Pollut 2019. In this current study, the authors attempted to establish a connection between 4-tOP and CIMT by analyzing the apoptotic microparticles. Although the study found that 4-tOP levels may negatively attenuate endothelial cell apoptosis and arteriosclerosis, the study is insufficient to conclude that 4-tOP has a positive impact on cardiovascular health, as various other approaches are required to validate the findings. However, the study may provide another role of harmful 4-tOP which may be useful for researchers.
There are some concerns that need to be addressed:
1. 4-tOP has been known to have numerous adverse effects on human health. What is the clinical application of the study?
2. As the study is done on the Taiwanese population, the authors can specify it in the Title of the manuscript.
3. To make it understandable to a broader audience, figure legends need detailed explanations.
4. Please describe what is (a) and (b) in Figure 1
5. Please include a clear protocol of how serum 4-tOP is measured in the main methodology. This will be helpful for reproducibility.
6. There are irrelevant texts incorporated in the manuscript. Please check lines 89 to 103.
There are minor grammatical errors that should be corrected.
Reviewer 2 Report
This study by Lin et al. aims to explore the potential inverse co-correlation between 4-tOP and apoptotic microparticles, suggesting that 4-tOP might contribute to improving endothelial cell (EC) apoptotic processes and potentially mitigate atherosclerosis. However, it's important to note that the presented data are solely observational, stemming from a human cohort study. While the authors mention that "if these associations are etiologic, the results suggest that 4-tOP might attenuate the apoptotic process," there is a lack of concrete data to support or even suggest this conclusion.
A significant aspect to consider is a prior study by the same authors demonstrating a negative correlation between 4-tOP and CIMT. Apoptotic microparticles have been established to correlate with CIMT, as evidenced in a previous study by the same authors (PMID: 27288966). Essentially, this current study put together the positive and negative correlations of these factors. To make this study relevant, it would be essential for the authors to demonstrate, at least in vitro, the potential of 4-tOP to modulate apoptotic microparticle formation. This could be done, for instance, by investigating the effects of 4-tOP in response to palmitate or other pro-apoptotic agents in ECs.
The title is misleading, as this paper doesn’t actually provide evidence of a mediating negative correlation between apoptotic microparticles and 4-tOP. Should be changed
The text from lines 89 to 103 appears to be related to formatting.
Round 2
Reviewer 2 Report
Thanks for the reply,
I still believe that the novelty of these findings are debatable, however the changes make it more in line with the data provided.